# How good are GPs at adhering to a pragmatic trial protocol in primary care? Results from the ADDITION-Cambridge cluster-randomised pragmatic trial

Michael Laxy,[1,2,3] Edward C F Wilson,[4] Clare E Boothby,[3] Simon J Griffin[3]

[1]Institute of Health Economics, Helmholtz Zentrum München, Neuherberg, Germany
[2]German Center for Diabetes Research, Neuherberg, Germany
[3]MRC Epidemiology Unit, University of Cambridge, Cambridge, UK
[4]Cambridge Centre for Health Services Research, University of Cambridge, Cambridge, UK

**Correspondence to**
Dr Edward C F Wilson;
ed.wilson@medschl.cam.ac.uk

## ABSTRACT

**Objective** To assess the fidelity of general practitioners' (GPs) adherence to a long-term pragmatic trial protocol.
**Design** Retrospective analyses of electronic primary care records of participants in the pragmatic cluster-randomised ADDITION (Anglo-Danish-Dutch Study of Intensive Treatment In People with Screen Detected Diabetes in Primary Care)-Cambridge trial, comparing intensive multifactorial treatment (IT) versus routine care (RC). Data were collected from the date of diagnosis until December 2010.
**Setting** Primary care surgeries in the East of England.
**Study sample/participants** A subsample (n=189, RC arm: n=99, IT arm: n=90) of patients from the ADDITION-Cambridge cohort (867 patients), consisting of patients 40–69 years old with screen-detected diabetes mellitus.
**Interventions** In the RC arm treatment was delivered according to concurrent treatment guidelines. Surgeries in the IT arm received funding for additional contacts between GPs/nurses and patients, and GPs were advised to follow more intensive treatment algorithms for the management of glucose, lipids and blood pressure and aspirin therapy than in the RC arm.
**Outcome measures** The number of annual contacts between patients and GPs/nurses, the proportion of patients receiving prescriptions for cardiometabolic medication in years 1–5 after diabetes diagnosis and the adherence to prescription algorithms.
**Results** The difference in the number of annual GP contacts (β=0.65) and nurse contacts (β=−0.15) between the study arms was small and insignificant. Patients in the IT arm were more likely to receive glucose-lowering (OR=3.27), ACE-inhibiting (OR=2.03) and lipid-lowering drugs (OR=2.42, all p values <0.01) than patients in the RC arm. The prescription adherence varied between medication classes, but improved in both trial arms over the 5-year follow-up.
**Conclusions** The adherence of GPs to different aspects of the trial protocol was mixed. Background changes in healthcare policy need to be considered as they have the potential to dilute differences in treatment intensity and hence incremental effects.
**Trial registration number** ISRCTN86769081.

### Strengths and limitations of this study

► Pragmatic trials aim to produce externally valid results for decision makers. If and to what extent pragmatic trial interventions are delivered to patients often remains unknown.
► This study describes the adherence of general practitioners to the ADDITION trial protocol (Anglo-Danish-Dutch Study of Intensive Treatment In People with Screen Detected Diabetes in Primary Care), and hence provides a unique insight about what we can expect in future long-term pragmatic studies in the primary care context, particularly in the context of policy and guideline changes.
► Analyses are based on a subsample of participants of the ADDITION-Cambridge trial conducted in the East of England. Therefore, the generalisability of results might be limited.

## BACKGROUND

Type 2 diabetes is an increasing public health problem associated with premature mortality and costly microvascular and macrovascular complications in terms of both reduced quality of life and financial burden, causing substantial economic pressure on healthcare systems and societies.[1–4]

Previous research has shown that intensive treatment of cardiovascular risk factors is an effective and cost-effective intervention for patients with long-standing diabetes or routinely diagnosed diabetes.[5–8] In contrast, little was known about the effectiveness and cost-effectiveness of intensive primary care-based treatment in patients in the early stages of the disease, such as screen-detected populations. The pragmatic cluster-randomised ADDITION (Anglo-Danish-Dutch Study of Intensive Treatment In People with Screen Detected Diabetes in Primary Care) trial (ISRCTN86769081) was one of the first

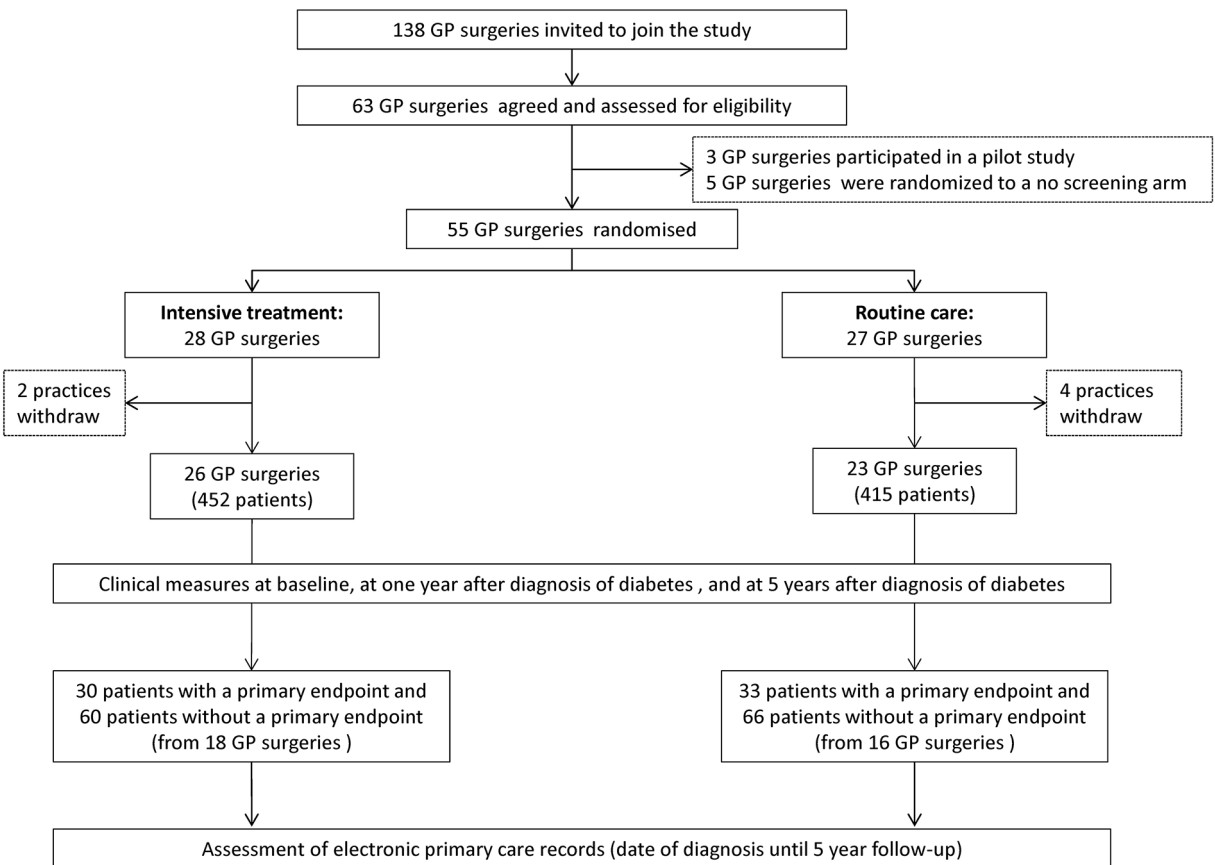

**Figure 1** Study design. GP, general practitioner.

studies addressing this important question.[9–11] Results showed that, compared with routine care, early intensive treatment modestly improved levels of cardiovascular risk factors, but did not significantly reduce the incidence of cardiovascular events, microvascular complications and cardiovascular/overall mortality over the 5-year study period.[12–14]

Pragmatic trials aiming to generate externally valid evidence in a real-world setting, such as ADDITION, always present uncertainties concerning the implementation of the planned interventions in daily practice. Unlike highly controlled efficacy trials in which compliance to an (simple, one-dimensional) intervention can (and must) be assured, the purpose of pragmatic trials is to assess the effectiveness of an (complex, multifactorial) intervention in routine settings. In the ADDITION-Cambridge trial, intensive treatment (IT) was compared with routine care (RC) for patients with screen-detected diabetes. IT in ADDITION was a multifactorial intervention including treatment targets and treatment algorithms that were more intensive than those in contemporary UK national treatment guidelines, as well as educational material for patients.[10 15–17] However, the degree to which protocol components were implemented into practice, and hence the degree to which more intensified treatment was actually provided to patients in the intervention arm, has remained unknown. Furthermore, potential changes in national treatment guidelines towards more intensive care, and the introduction of the pay for

performance system in England within the national Quality and Outcomes Framework (QOF),[18 19] are likely to have improved routine care and may have diluted the difference in treatment intensity between the study arms over time.[20]

Beyond improving understanding of the results of the ADDITION-Cambridge study, knowledge about whether and how the intervention was actually delivered in practice can inform future pragmatic trials in relation to barriers to protocol adherence, and the difference in treatment intensity that can be expected in a primary care-based pragmatic trial in the context of background policy changes.

The objective of this study was therefore to describe the adherence of general practitioners (GPs) to the trial protocol and to compare the intensity of care delivered to patients with screen-detected diabetes between the trial arms.

## METHODS
### Study design
The ADDITION-Cambridge study protocol has been published elsewhere.[10] In brief, ADDITION-Cambridge is part of the ADDITION-Europe trial, which consisted of two phases: a screening programme and a pragmatic, cluster-randomised trial comparing the effect of early IT versus RC on 5-year cardiovascular risk in patients with screen-detected type two diabetes mellitus.[9] The primary endpoint was a composite of cardiovascular morbidity

**Table 1** Criteria for the initiation of glucose-lowering, blood pressure-lowering, lipid-lowering and platelet-inhibiting (aspirin) medication according to the trial protocol (IT arm) and national guidelines (RC arm)*

| | Glucose-lowering therapy | Blood pressure-lowering therapy | Lipid-lowering therapy | CVD risk-lowering aspirin therapy |
|---|---|---|---|---|
| Routine care (RC) | – if HbA$_{1c}$ ≥7%† | – if BP ≥160/100 <br> – if 140/80 mm Hg ≤BP < 160/100 mm Hg and either prevalent CVD or 10-year CHD risk ≥15% <br> (ACE inhibitors, ARBs, beta-blockers or diuretics as first choice) | – if total cholesterol ≥5 mmol/L or triglycerides ≥2.3 mmol/L <br> – if prevalent CVD or 10-year CHD risk ≥15% | – if prevalent CVD or 10-year CHD risk ≥15% |
| Intensive treatment (IT) | – if HbA$_{1c}$ ≥6.5% | – if ≥120/80 mm Hg or prevalent CVD <br> (ACE inhibitors/ARBs as first choice) | – if total cholesterol ≥3.5 mmol/L | – independent of risk profile |

This figure does not claim to comprehensively describe the national treatment algorithms from the year 2002 or the detailed ADDITION trial protocol. It only highlights the differences in criteria for the initiation of drug therapy between IT and RC and does not account for possible contraindications.

*Criteria are based on the national treatment guidelines from 2002[15–17] and the ADDITION trial protocol.[10]

†A range of 6.5%–7.5% was mentioned. Consequently, the arithmetic mean of the borders (7%) was used as threshold.

ADDITION, Anglo-Danish-Dutch Study of Intensive Treatment In People with Screen Detected Diabetes in Primary Care; ARB, angiotensin receptor blockers; BP, blood pressure; CHD, coronary heart disease; CVD, cardio-vascular disease; HbA$_{1c}$, glycated haemoglobin.

and mortality (cardiovascular death, non-fatal myocardial infarction, non-fatal stroke, non-traumatic amputations and revascularisations).

## Study population

For ADDITION-Cambridge, 33 539 eligible individuals were invited to stepwise screening. Individuals eligible for screening were people registered at one of the participating general surgeries around Cambridge, aged 40–69

**Table 2** Baseline characteristics of the used subsample of ADDITION-Cambridge

| | Intensive treatment | Routine care |
|---|---|---|
| N | 82 | 91 |
| Female sex, n (%) | 30 (36.6) | 30 (30.3) |
| Caucasian ethnicity, n (%) | 77 (93.9) | 96 (97) |
| Age, mean (SD) | 61.87 (7.28) | 62.01 (6.81) |
| BMI (kg/m$^2$), mean (SD) | 33.6 (5.6) | 33.8 (5.9) |
| Total cholesterol (mmol/L), mean (SD) | 5.47 (1.12) | 5.46 (1.22) |
| HDL cholesterol (mmol/L), mean (SD) | 1.16 (0.32) | 1.2 (0.31) |
| Systolic blood pressure (mm Hg), mean (SD) | 143 (20.8) | 143.8 (22.2) |
| HbA$_{1c}$(%), mean (SD) | 7.84 (2.09) | 7.27 (1.59) |

ADDITION, Anglo-Danish-Dutch Study of Intensive Treatment In People with Screen Detected Diabetes in Primary Care; BMI, body mass index; HbA$_{1c}$, glycated haemoglobin; HDL, high-density lipoprotein; N, number of individuals included in the analysis sample.

years, not known to have diabetes and with a diabetes risk score of >0.17 (corresponding to the top 25% of the population distribution). The risk score included age, sex, body mass index, steroid and antihypertensive medication, as well as smoking and family history.[21] Exclusion criteria were assessed by the potential participant's GP. Patients with severe illness with a life expectancy of less than 12 months, those with psychological or psychiatric disorders that might invalidate informed consent, and those who were housebound, pregnant or breast feeding were excluded from the study. Eight hundred and sixty-seven eligible patients (from n=49 surgeries) with screen-detected diabetes participated in the pragmatic primary care-based intervention trial. Written informed consent was obtained from all participants. This trial is registered as ISRCTN86769081.

Due to the high expenses of assessing and extracting data from electronic primary care records, it was decided in the planning phase of the ADDITION-Cambridge study that only the records of a subset of the study will be assessed. It was decided that the records of participants with a primary endpoint within the 5 years of follow-up plus the records of two random participants without a primary endpoint from the same GP surgery will be accessed. Consequently, the records of 63 participants with a primary endpoint (30 from the IT arm and 33 from the RC arm) and of 126 participants without a primary endpoint (60 from the IT arm and 66 from the RC arm) were collected. This selection procedure led in total to a subsample of 189 participants (IT: n=90 patients, RC: n=99 participants) from 34 surgeries (IT: 18 GP surgeries, RC: 16 GP surgeries). The study design is illustrated in detail in figure 1.

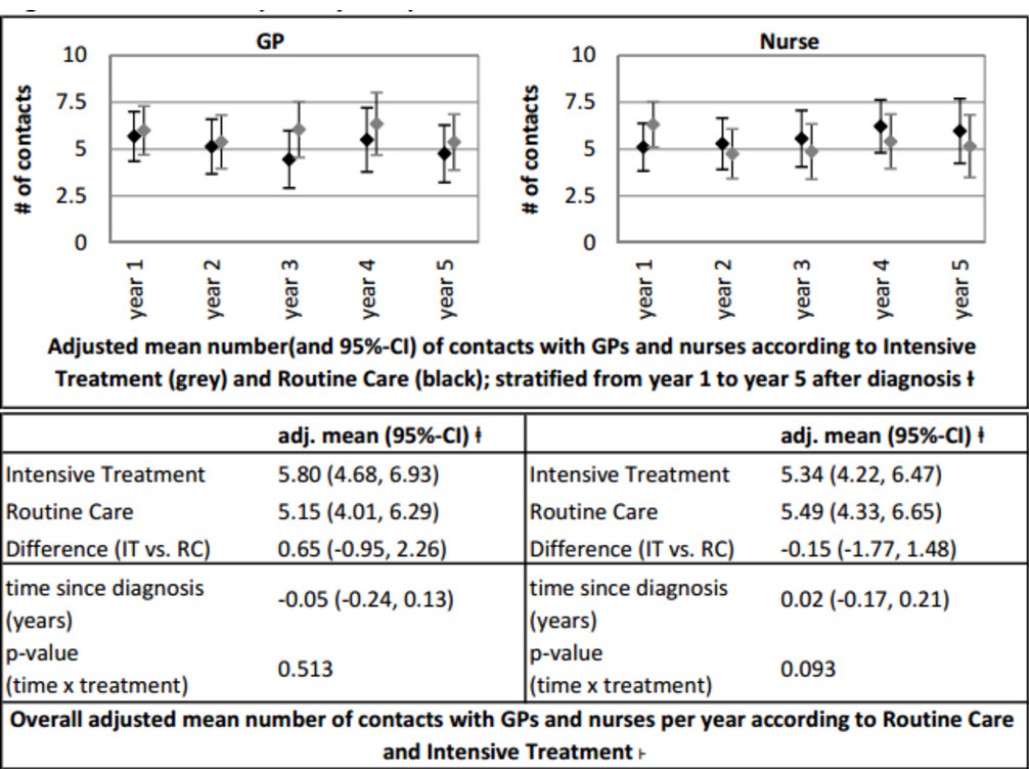

ꝉ stratified linear regression models with a main effect for the intervention; adjusted for sex and age of diagnosis; accounted for patients being clustered within GP practices

⊦ overall linear regression models with a main effect for the intervention and for time since diagnosis and an interaction term between intervention and time; adjusted for sex and age of diagnosis; accounted for patients being clustered within GP practices and observations being clustered in patients

‡ n=827 observations (n=169 in year 1, n=167 in year 2, n=168 in year 3, n=164 in year 4 and n=159 in year 5)

**Figure 2** Contact with primary care professionals.

### IT and RC

Patients were treated according to the treatment allocation of their surgery. In the RC arm patients received diabetes care through the National Health Service according to current UK guidelines and recommendations.[15–17] In the IT arm additional features were added to current RC:

a. Surgeries received funding for three additional 10 min GP consultations and three additional nurse consultations per year in the first 3 years after diagnosis.

b. Treatment algorithms were introduced along with underlying evidence demonstrating positive effects on cardio-vascular disease (CVD) risk factors among patients with type 2 diabetes. In the IT arm therapy with glucose-lowering medication was indicated if glycated haemoglobin ($HbA_{1c}$) ≥6.5%; ACE inhibitors/angiotensin receptor blockers (ARBs) if blood pressure (BP) ≥120/80 mm Hg; statins if cholesterol ≥3.5 mmol/L; and aspirin for all patients independent of their risk factor levels (assuming that patients had no contraindications). The thresholds for treatment initiation for glucose-lowering, BP-lowering and lipid-lowering medication and for aspirin therapy in both the IT arm (based on the trial protocol[10]) and the RC arm (based on national guidelines[15–17]) are summarised in table 1.

c. 3. Practice teams received theory-based educational materials to hand over to the patients, aiming to provide a shared framework for the management of their disease. Furthermore, GPs were advised to refer patients to a dietitian, and patients were encouraged through their GPs and nurses to increase their physical activity, to avoid excessive alcohol intake, to lose weight, to stop smoking, to adhere to medication and to self-monitor blood glucose if given a glucometer by their GP.

Intensive treatment was promoted to participating surgeries by practice-based educational meetings with GPs and nurses. This included initial practice-based academic detailing conducted by a diabetologist and an academic GP to introduce treatment algorithms, and two interactive practice-based feedback sessions (approximately 6 and 14 months after the initial education session) to support and monitor treatment delivery.

### Measures of treatment intensity

Information on the intensity of delivered care was extracted from the electronic primary care records of participating

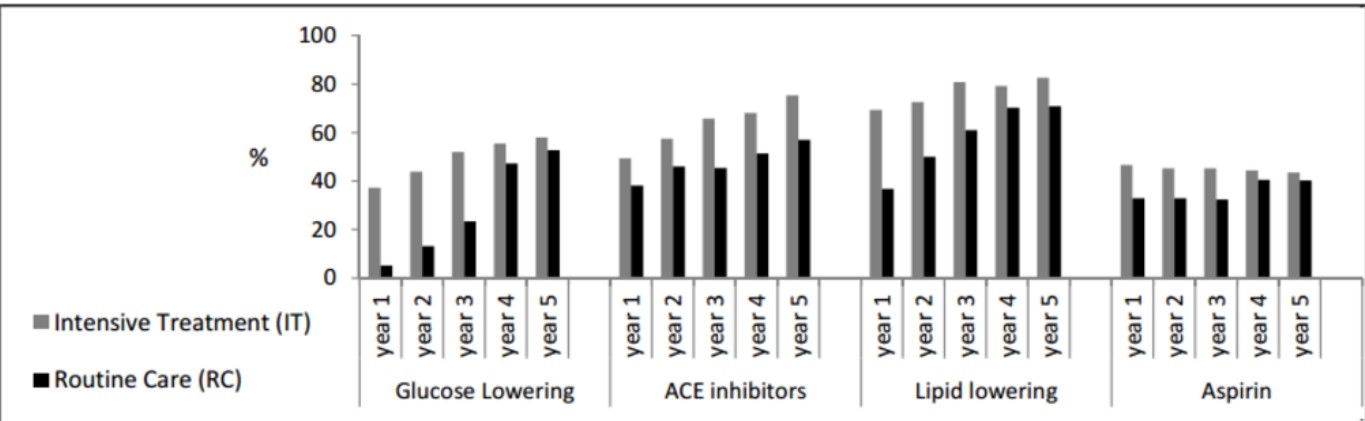

Proportion of patients receiving at least 4 prescriptions according to Intensive Treatment (grey) and Routine Care (black), stratified for year 1 - 5 after diagnosis

| Stratified by year ‡ | OR (95%-CI) ‡ | OR (95%-CI) ‡ | OR (95%-CI) ‡ | OR (95%-CI) ‡ |
|---|---|---|---|---|
| Year 1 (IC vs. RC) | 10.89 (3.53, 33.56) | 1.57 (0.73, 3.37) | 4.00 (1.95, 8.20) | 1.67 (0.72, 3.85) |
| Year 2 (IC vs. RC) | 5.88 (2.51, 13.80) | 1.60 (0.82, 3.09) | 2.63 (1.31, 5.26) | 1.66 (0.72, 3.86) |
| Year 3 (IC vs. RC) | 3.78 (1.76, 8.10) | 2.34 (1.18, 4.64) | 2.63 (1.15, 6.01) | 1.60 (0.62, 4.09) |
| Year 4 (IC vs. RC) | 1.42 (0.73, 2.76) | 2.06 (1.02, 4.14) | 1.57 (0.68, 3.63) | 1.16 (0.37, 3.61) |
| Year 5 (IC vs. RC) | 1.23 (0.62, 2.42) | 2.66 (1.14, 6.21) | 1.99 (0.88, 4.53) | 1.22 (0.43, 3.50) |
| **Year 1-5** ⊦ | **OR (95%-CI) #** | **OR (95%-CI) #** | **OR (95%-CI) #** | **OR (95%-CI) #** |
| Overall (IC vs. RC) | 3.27 (1.81, 5.93) | 2.03 (1.13, 3.65) | 2.42 (1.3, 4.51) | 1.41 (0.61, 3.24) |
| Time since diagnosis (per year) | 1.61 (1.42, 1.83) | 1.25 (1.12, 1.39) | 1.33 (1.18, 1.5) | 1.04 (0.93, 1.15) |
| p-value (time x treatment) | <.0001 | 0.331 | 0.131 | 0.220 |

Odds Ratio of having received at least 4 prescriptions per year IT vs. RC (reference)

‡ stratified logistic regression models with a main effect for the intervention; adjusted for sex and age of diagnosis; accounted for patients being clustered within GP practices

⊦ overall logistic regression models with a main effect for the intervention and for time since diagnosis and an interaction term between intervention and time; adjusted for sex and age of diagnosis; accounted for patients being clustered within GP practices and observations being clustered in patients

‡ n=151 in year1, n=149 in year 2, n=150 in year 3, n=146 in year 4, and n=141 in year 5

# n=737 observations (n=151 in year1, n=149 in year 2, n=150 in year 3, n=146 in year 4, and n=141 in year 5)

**Figure 3** Medication intensity.

patients from the date of the diabetes diagnosis until December 2010 by a researcher blind to the GP surgery study group allocation. These files recorded the date and type of delivered services, including consultations with primary care health professionals, prescribed medications and laboratory measurements/tests. For the analysed trial population more than 80 000 observations were available in the first 5 years after diagnosis. Clear text functions were used and algorithms were derived to classify the obtained information. Ambiguous observations were screened and coded by hand. Anatomic Therapeutic Chemical codes were assigned to drugs to categorise medication classes. The intensity of care indicators was defined as follows:

### Contact with healthcare professionals
The annual number of contacts between patients and GPs (including GP partners, GP principals, GP associates and out-of-hours doctors) and nurses (including practice nurses, nurse practitioners and nurse specialists). This included all contacts as we were unable to distinguish those related to diabetes alone.

### Medication
Continuous treatment (≥4 prescriptions annually) with glucose-lowering drugs (metformin, sulfonylurea, thiazolidinedione, insulin, other glucose-lowering drugs), ACE-inhibiting drugs (ACE inhibitors or ARBs), lipid-lowering drugs (statins, other cholesterol-lowering drugs) or aspirin.

### Monitoring of risk factor levels
Regular monitoring of glycaemic control (≥2 $HbA_{1c}$ tests per year), lipid profile (≥1 cholesterol test per year) and kidney function (≥1 urine albumin-creatinine ratio (UACR) test per year).[15–17]

### Statistical analyses
We analysed the difference in treatment intensity within the first 5 years from date of diagnosis. The study period

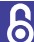

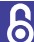

**Figure 4** Medication adherence.

was subdivided into five annual intervals representing year 1 (day 1–day 365) to year 5 (day 1460–day 1825) from diagnosis. Sixteen patients whose electronic primary care records did not contain information for at least one entire year were excluded from the analysis, resulting in an analysis sample of 173 patients from 34 GP surgeries with a mean cluster size of 5 patients (IT: 82 patients from 18 surgeries, RC: 91 patients from 16 surgeries). Due to non-availability of data, surgery changes and deaths, the total number of complete observed patient-years over the follow-up period was 827 for contact with healthcare professionals and monitoring and 737 for prescriptions.

We applied linear regression models separately for years 1–5 in order to analyse the difference in the number of contacts with GPs and nurses for each individual year. A multilevel linear regression model accounting for repeated observations (years 1–5) within patients was applied to test the overall difference in the number of annual contacts between the study arms over the 5-year study period. This model included an interaction term between the year since diagnosis and the treatment to capture any time–treatment interactions.

In parallel with the linear regression models for the frequency of contacts with healthcare professionals,

logistic regression models were applied to assess the likelihood of receiving continuous medication (≥4 prescriptions annually). In a secondary analysis, we also examined the likelihood of receiving regular monitoring of glycaemic control, lipid profile and kidney function and the likelihood of seeing a dietitian.[15–17]

Linear and logistic regression models were adjusted for age and sex and accounted for patients being clustered into surgeries (two-level model for stratified analyses and three-level models for overall analyses). As the non-random selection of the analysed subsample does not exactly represent the study population, we tested in a sensitivity analysis if the introduction of a weighting factor (inverse probability of being included in the study based on the status of having a primary endpoint) has an impact on the results. We also altered the thresholds for the definition of 'continuous' medication (from 4 to 2, 6 and 12 prescriptions) to assess the sensitivity towards these threshold definitions. To assess the sensitivity to missing data, we further refitted the analyses to a regression-based multiple-imputed (n=10 imputations) data set (n=189 patients). Statistical analyses were performed with SAS V.9.3 using the GLIMMIX, MI and MIANALYZE procedures.

To gain a more detailed insight into the pattern of GPs' adherence to treatment algorithms, we further extracted clinical information including HbA$_{1c}$, BP, cholesterol, triglycerides, prevalent CVD (defined as myocardial infarction or stroke) and 10-year modelled coronary heart disease (CHD) risk (using the United Kingdom Prospective Diabetes Study (UKPDS) risk engine V.2) from the baseline, year 1 and year 5 examinations of the ADDITION study. Missing clinical values were imputed by the methods of last observation carried forward and first observation carried backwards to avoid shrinkage of the sample size. We calculated the proportion of patients who should have received medication, that is, the proportion of patients whose clinical values exceeded the thresholds referred to in the trial protocol[10] and the national guidelines[15–17] (p (clinical value ≥threshold)) and the proportion of patients who actually received at least one prescription in a time frame of 3 months after the date of the laboratory measurement (p (number of prescriptions ≥1)) (table 1). We finally defined the adherence of GPs to the trial protocol/ national guidelines descriptively as the proportion of patients who receive at least one prescription, out of those patients whose clinical values exceed the thresholds (p (number of prescriptions ≥1) | (clinical value ≥threshold)).

## RESULTS
### Baseline sample characteristics
Characteristics of the sample at baseline are shown in table 2. The mean age of the sample was 62 years, 34% were female and 96% Caucasian. The biomedical characteristics of the comparison arms were balanced. No differences were observed between the full sample (n=189) and the analysis sample (n=173).

### Contact with healthcare professionals
The adjusted mean number of annual GP and nurse contacts is graphically illustrated in figure 2. We found no difference in the mean annual number of contacts with GPs (IT: 5.80 vs RC: 5.15, β=0.65 [95% CI −0.95 to +2.26] or nurses (IT: 5.34 vs RC: 5.49, β=−0.15 [95% CI −1.77 to +1.48]) and no statistically significant trend over time.

### Medication
The proportion of GPs who regularly prescribed (≥4 times annually) glucose-lowering and cardioprotective drugs and ORs for the likelihood of regular prescriptions are shown in figure 3.

GPs in the IT arm were 3.27 [95% CI 1.81 to 5.93] times more likely to regularly prescribe glucose-lowering medications compared with GPs in the RC arm. However, this difference diminished over the follow-up period as more patients in the RC arm were also prescribed medication. Patients in the IT arm also had a greater chance of being prescribed lipid-lowering medication (OR=2.42 [1.30 to 4.51]) and ACE-inhibiting drugs (OR=2.03 [1.13 to 3.65]), which were, in contrast to routine care guidelines, the first choice BP-lowering drug according to the trial protocol. But no significant difference was observed between the trial arms for the category of BP-lowering drugs as a whole (including beta-blocker, diuretics and so on) (OR=1.41 [0.71 to 2.80]) (see online supplementary appendix 1). No significant difference was observed between the trial arms for prescription of aspirin. Overall in both treatment arms, the likelihood of patients receiving glucose-lowering, ACE-inhibiting and lipid-lowering medications increased from diagnosis to 5-year follow-up.

### Monitoring of risk factors
The proportion of patients receiving regular HbA$_{1c}$ tests (≥2 annually, 45% of patients), lipid tests (≥1 annually, 55% of patients) and UACR tests (≥1 annually, 75% of patients) was low. No significant difference was observed between the treatment arms (HbA$_{1c}$ tests: OR=1.56 [0.63 to 3.83], lipid tests OR=1.53 [0.51 to 4.60] and UACR test: OR=0.82 [0.34 to 1.98]) (see online supplementary appendix 1).

### Sensitivity analysis
Analyses of multiple-imputed data sets led to qualitatively and quantitatively similar results. Also the introduction of a weighting factor to account for non-random patient selection yielded comparable results. Using different thresholds for the definition of 'continuous medication' showed that the results for glucose-lowering and lipid-lowering medications were not sensitive to threshold

definitions. However, increasing the threshold number for lipid-lowering drugs attenuated the respective OR considerably (online supplementary appendix 2).

## Adherence to prescription algorithms

The proportions of patients who should have received medication according to national guidelines and the ADDITION trial protocol and the proportions of patients who actually received a prescription within 3 months following the assessment of biomedical data are presented in column 1 and column 2 of figure 4: The black part in column 2 represents the proportion of patients who received a prescription and whose clinical values exceeded the thresholds for medication prescription, and the framed white part represents the proportion of patients who received medication, although clinical values did not exceed the thresholds. Adherence to the prescription algorithms, that is, the proportion of patients who received at least one prescription out of those patients whose clinical values exceeded the thresholds (p (number of prescriptions $\geq1$) | (clinical value $\geq$threshold)) is shown numerically in the lower part of figure 4.

Due to tighter algorithms in the trial protocol (IT arm) than in the national guidelines (RC arm), more patients in the IT arm were eligible for glucose-lowering, BP-lowering and aspirin therapy than in the RC arm. However, despite lower cholesterol thresholds in the IT arm compared with the RC arm, treatment with lipid-lowering medication was indicated in almost equal proportions of patients in the two treatment arms.

### Glucose-lowering drugs

In the first year, the adherence to the treatment algorithm was generally low, but considerably higher in the IT arm than in the RC arm. At year 5, 73% of patients in both treatment arms with an $HbA_{1c} \geq$ threshold level received a prescription.

### BP-lowering/ACE-inhibiting drugs

In the IT arm, adherence to the guideline for prescription of ACE-inhibiting medication increased from 41% at baseline to 77% at year 5. In the RC arm, guideline adherence for prescription of any BP-lowering medication increased from 55% at baseline to 94% at year 5, and 'prescription adherence' to ACE-inhibiting medication (ACE inhibitors were not mentioned in the guidelines to be the first-line treatment in RC) increased from 28% at baseline to 64% at year 5 (not shown). Of note, a large proportion of patients in the RC arm with BP levels below the threshold were prescribed BP-lowering medication.

### Lipid-lowering drugs

Adherence to the treatment algorithms increased in both treatment arms and was consistently better in the IT arm. At year 5, most patients with clinical values greater than the threshold levels were treated (IT arm 93%, RC arm 81%).

### Aspirin

The adherence to the trial protocol/guidelines was low; less than 50% of eligible patients in both treatment arms received aspirin.

# DISCUSSION
## Summary

ADDITION is a large pragmatic primary care-based trial aiming to promote intensive multifactorial treatment of patients with screen-detected diabetes by GPs. Utilising electronic primary care records of patients, this study shows that GPs in the IT arm did not see their patients more often, but were more likely to regularly prescribe metabolic and cardioprotective drugs. Generally, GPs' adherence to prescription algorithms increased substantially in both trial arms over the 5-year follow-up period. The substantial time–treatment interaction for prescription of glucose-lowering medication indicate that background changes in routine care might have diluted the difference in treatment intensity over time.

## Contextual frame

Pragmatic ('effectiveness') trials seek to produce externally valid results in order to inform the process of decision-making by policy makers.[22–25] However, unlike in explanatory ('efficacy') trials, adherence to protocol is rarely tightly monitored and the degree to which the intervention is implemented often remains uncertain. In the case of non-statistically significant results, this begs the question whether the intervention is per se not efficacious in the tested (heterogeneous) population, or whether the intended difference in treatment intensity was not big enough to detect any effects in the given sample size.

Lack of a difference in the intensity of treatment can be due to different reasons. First, adherence of responsible healthcare professionals to the protocol might be low due to limited motivation, insufficient resources or lack of interest in the ongoing trial. To tackle this issue, in ADDITION-Cambridge, a detailed trial protocol was specified and the implementation of the protocol elements was incentivised by additional monetary resources and supported by an initial practice-based academic and two interactive feedback sessions.[10]

Second, treatment delivered in everyday practice might differ from both guidelines and what happens in research-active practices. Not considering actual practice in routine care can result in intervention plans that fail to induce treatment differences between the trial arms. The choice of suitable interventions is therefore particularly challenging in multinational trials like ADDITION, where guidelines or daily practice in countries might differ, but a certain degree of intervention homogeneity is warranted.[9]

Third, policy changes, such as changes in the remuneration system and modifications in treatment guidelines, can intensify routine care, thus potentially diluting

differences between the intervention and routine care arm. Long-term trials such as ADDITION are particularly susceptible to such influences. Between 2003 (~start of the study) and 2008/2009 (~end of the 5-year analysis period) in the UK, no new national diabetes treatment guidelines were released. However, in 2004 the QOF with its pay for performance system was launched[18] and extended in the following years. The QOF incentivised fulfilment of basic quality of care indicators by monetary resources and may have improved the quality of care for patients with various conditions, including diabetes.[20 26]

### Principal findings

Our study shows that although surgeries in the IT arm received monetary resources for additional consultations, GPs and nurses did not see their patients more often, nor were they more likely to perform regular $HbA_{1c}$, lipid or UACR tests. This result might be explained by the fact that the patients in the RC arm already saw their GP/nurse on average five to six times a year, which is more than the average ~4 GP and ~2.5 nurse contacts per year for the general UK population.[27] Therefore the GPs (and indeed the patients) may have felt that this was sufficient to adequately monitor the condition. It also shows that monetary incentives might help to convince a reasonable number of surgeries to participate in long-term extensive trials such as ADDITION (46% of contacted surgeries agreed to join the study), but that financial incentives might not be successful in motivating GPs to further increase treatment intensity if it is already at a high level.[10] Qualitative interviews with the GPs about their perspectives on the intervention, as conducted in the screening phase of the ADDITION study,[28] would have been a valuable add-on to address this question. In contrast, our results indicate that the education sessions and feedback audits had a positive impact on the protocol adherence of GPs, as general adherence to the treatment algorithms in the IT arm was higher than adherence to the national guidelines in the RC arm. This finding supports previous research that feedback loops can help to maximise guideline adherence in primary care.[29 30]

According to the clinical thresholds outlined in the trial protocol and the national guidelines, more patients in the IT arm than in the RC arm were eligible to receive glucose-lowering, BP-lowering and platelet-inhibiting drugs (figure 4). This suggests that the ADDITION intervention was designed at an appropriate level for the context, as even with a hypothetical prescription adherence of 100%, patients in the IT arm should have received more intensive treatment than patients in the RC arm.

Notably, a very high proportion of patients in the RC arm already received BP-lowering medication at baseline, although in many cases their BP levels did not exceed thresholds. The finding of high BP-lowering prescription prevalence probably results from the fact that treatment with BP-lowering medication was part of the risk score used to identify high-risk individuals eligible for diabetes screening in the first phase of the

ADDITION trial.[10] There could be two reasons why many of the patients who received BP-lowering prescriptions had no apparent clinical indication for treatment. On the one hand, these patients might have previously had uncontrolled BP levels, but treatment with BP-lowering medication brought their BP under control. On the other hand, it is possible that the daily practice for BP control at this time was already much stricter than recommended by the guidelines. Independent of its origin, the initially high prevalence of BP-lowering medication in both trial arms might be the reason why we did not observe a difference in the proportion of patients prescribed BP-lowering drugs. Consequently, the observed difference in ACE-inhibiting drugs may be due to GPs switching from diuretics or beta-blockers to ACE-inhibiting drugs, as recommended by the trial protocol.

The low adherence to recommendations concerning aspirin therapy observed in both trial arms is interesting, as this prescription behaviour could be interpreted as a general scepticism among GPs (and perhaps patients) towards the weak evidence of benefits of aspirin therapy for primary prevention of cardiovascular disease.[6] The results of subsequent large trials justify such scepticism.[31 32] Alternatively, some patients may have obtained aspirin from the pharmacy without a prescription without this being noted in the electronic medical record.

Except for aspirin, adherence to prescription algorithms increased substantially over the follow-up period. We assume that this finding is triggered by the progression and duration of the disease and by general improvements in the overall quality of care over time, independent of disease progression.[33] The significant interaction between 'treatment' and 'time since diagnosis' for glucose-lowering medication indicates changing treatment patterns in the RC arm, which might be triggered by policy changes, like QOF. However, due to methodological limitations (covariate collinearity, power problems in stratified models), this question could not be adequately addressed with the available data.

### Implications for the planning of future pragmatic trials

This study shows that the successful implementation of a pragmatic trial in primary care is possible, but there are issues that need to be considered, namely (1) a high standard of care in control GP surgeries that questions the need for further intensification, (2) treatment of patients in the RC arm that did not reflect the national guidelines, and (3) background policy changes affecting quality of routine care. These issues need to be identified, considered and addressed when designing a pragmatic study or rolling out an intervention comprehensively.[23 24 34] The results further underline the potential importance of standard good practice in (pragmatic) trials. Methods such as initial academic detailing and repeated feedback sessions may be of great importance for the overall success of the study.[24 35] In this context, more qualitative or

quantitative implementation research may help to identify and test strategies that affect the adherence of healthcare professionals (and patients).[36]

Ideally, pragmatic trials of complex interventions should, if possible, be designed in a way that allows evaluation of the adherence of healthcare professionals to the trial protocol and of patients to the chosen treatment regimen. This study shows that the use of electronic primary care records is a promising approach to assess the adherence of GPs. The obtained data are also useful for health economic research. In this particular example, the new primary care data can be used to update a previous analysis to reduce uncertainty in the cost-effectiveness of the intervention,[37 38] a method consistent with an iterative approach to research and adoption decisions.[39–41]

## Implications for the interpretation of trial results

Intensified prescription algorithms were well implemented into practice. We found that prescription with glucose-lowering, ACE-inhibiting and lipid-lowering drugs was higher in the IT arm. The expected treatment effect resulting from this difference in medication could be interpreted as an area under the curve issue: The combination of the magnitude and the duration of the treatment difference can be expected to be the crucial driver of long-term effects. The extended follow-up of the UKPDS trial, which aimed to reduce diabetes-related complications through tighter glucose and BP control, has shown that after the termination of the intervention, between-group differences in laboratory measurements disappeared.[42–45] However, the reductions in risk of microvascular and macrovascular complications persisted (or increased) for patients who had received tight glucose control, but not for patients who had received tight BP control.[42 43] In ADDITION we observed a small but significant improvement in HbA$_{1c}$, BP and cholesterol levels in the IT arm and a non-significant reduction in risk of the composite CVD endpoint (Relative Risk =0.83, p=0.12) over a 5-year time period.[14] This study shows that the proportion of patients receiving glucose-lowering drugs in each arm had equalised at the end of the 5-year observation period, suggesting that the differences in glycaemic control might disappear in the subsequent years. However, as a substantially greater proportion of patients in the IT arm received ACE-inhibiting and lipid-lowering drugs, it can be assumed that differences in BP and lipids might be sustained. If between-group differences in treatment for BP and lipids diminish, so will the levels of risk factors. However, the CVD risk may remain lower due to legacy effects of earlier reductions in glucose and cholesterol. Given that the number of events will also increase over time, it may be that the ADDITION intervention will show a statistically significant effect in the long term; the 10-year follow-up of ADDITION will quantify the long-term effect of relatively small differences in treatment and risk factors observed in the first 5 years after diagnosis of diabetes by screening.[14]

## Strengths and limitations

To our knowledge, this is one of the first studies to comprehensively analyse the adherence of GPs to a pragmatic trial protocol in primary care. In contrast to self-reported information from patients, electronically stored primary care records provide a high degree of detail about all GP-based primary care services delivered to patients and are less susceptible to recall bias.[46] Through the linkage of clinical information from the trial measurements with information on prescriptions from the electronic primary care records, it was further possible to comprehensively describe and analyse the prescription adherence of GPs to the trial protocol and to national guidelines.

However, we only had data from a subsample of the ADDITION-Cambridge trial cohort with an over-sampling of patients with a primary event during the follow-up period. As our weighted sensitivity analyses showed that this issue did not affect the results, the findings of this study are likely to be generalisable to the sample of GP surgeries who participated in the ADDITION trial. Nevertheless, the generalisability of results to average GP surgeries in the UK might be quite limited. In the experience of the authors, the practices that take part in research tend to be more organised and deliver better quality routine care than those declining to participate. This might lead to ceiling effects for interventions, that is, it appears to be hard to induce a difference in treatment intensity between RC and a more intensive treatment regimen.

Another limitation is that in our assessment of prescription adherence, we did not take into account possible contraindications for medications as well as patients' views, and analysed the data from a rather non-situational, disease-orientated perspective.[47 48] Shared decision making between the GP and the patient might reasonably lead to decisions that deviate from those in the protocol (and national guidelines). We therefore do not know if patients or GPs were the main determinants of protocol non-adherence. It is possible that patients did not agree to start medication or to come to the surgery more often. To completely understand the adoption of the intervention, the patient's role also needs to be taken into account, which was impossible with the chosen approach. Also, with the given data we could not evaluate the fidelity of GPs handing over the educational materials to study participants, which were also part of the intervention.

Finally, although the accuracy of primary care records for GP-based services is known to be quite high, particularly for prescribed medication and laboratory tests, the handling, merging and extraction of free text data from numerous observations (~80 000) originating from different IT format systems are challenging and validation was not undertaken.[46] Consequently, it is possible that a small proportion of services might be misclassified, resulting in non-differential bias.

## CONCLUSION

This study demonstrates that the successful implementation of long-term pragmatic trials in primary care is possible, but there are many obstacles especially during periods of significant change in routine care. The retrospective analyses of the electronic primary care records of participants in the ADDITION-Cambridge trial show that intensive treatment was fairly well implemented into practice, suggesting that positive effects on cardiovascular morbidity and mortality might be expected in the long term. Where possible, data needed to evaluate the fidelity of stakeholders to trial protocols should be collected routinely in future pragmatic trials as this information is invaluable for the interpretation of study results and for the planning of future studies.

**Acknowledgements** We gratefully acknowledge the contribution of all participants, practice nurses and general practitioners in the ADDITION-Cambridge study (a full list of participating practices is given below), and Kit Coutts and Dr Rebecca Simmons for their contribution to data collection. We also acknowledge the contribution of the trial steering committee (Professors Nigel Stott (Chair), John Weinman, Richard Himsworth and Paul Little) and Professor Jane Armitage and Dr Louise Bowman who adjudicated the endpoints. The Primary Care Unit and the Medical Research Council Epidemiology Unit at the University of Cambridge jointly coordinated the study. Aside from the authors, the ADDITION-Cambridge study team has included Rebecca Abbott, Amanda Adler, Judith Argles, Gisela Baker, Rebecca Bale, Ros Barling, Daniel Barnes, Mark Betts, Sue Boase, Sandra Bovan, Gwen Brierley, Ryan Butler, James Brimbicombe, Parinya Chamnan, Sean Dinneen, Pesheya Doubleday, Justin Basile Echouffo-Tcheugui, Sue Emms, Mark Evans, Tom Fanshawe, Francis Finucane, Philippa Gash, Julie Grant, Wendy Hardeman, Robert Henderson, Susie Hennings, Muriel Hood, Garry King, Ann-Louise Kinmonth, Georgina Lewis, Christine May Hall, Joanna Mitchell, Richard Parker, Nicola Popplewell, A Toby Prevost, Emanuella De Lucia Rolfe, Richard Salisbury, Lincoln Sargeant, Rebecca Simmons, Stephen Sharp, Megan Smith, Stephen Sutton, Nicholas Wareham, Liz White, Fiona Whittle and Kate Williams. We also wish to thank the Cambridge University Hospitals NHS Foundation Trust Department of Clinical Biochemistry and the NIHR Cambridge Biomedical Research Centre Core Biochemistry Assay Laboratory for carrying out the biochemical assays and the following groups within the MRC Epidemiology Unit: data management (Adam Dickinson), information technology (Iain Morrison, Rich Hutchinson), technical (Matt Sims) and field epidemiology (Paul Roberts, Kim Mwanza, James Sylvester, Gwen Brierley, Jaimie Taylor). ADDITION-Cambridge practices: Acorn Community Health Centre, Arbury Road Surgery, Ashwell Surgery, Birchwood Surgery, Bridge Street Medical Centre, Brookfields & Cherry Hinton, Broomfields, Buckden Surgery, Burwell Surgery, Cambridge Surgery, Cedar House Surgery, Charles Hicks Centre, Chequers Lane Surgery, Clarkson Surgery, Cornerstone Practice, Cornford House Surgery, Cottenham Surgery, Cromwell Place Surgery, Dr Smith and Partner (Cambridge), East Field Surgery, Ely Surgery, Freshwell Health Centre, George Clare Surgery, Great Staughton Surgery, Harston Surgery, Health Centre (Eaton Socon), Hilton House, John Tasker House, Lensfield Medical Practice, Manea Surgery, Mercheford House, Milton Surgery, Nene Valley Medical Practice, Nevells Road Surgery, New Roysia Surgery, Northcote House Surgery, Nuffield Road Medical Centre, Orchard Surgery, Orchard House Surgery, Orton Medical Practice, Park Medical Centre, Paston Health Centre, Peterborough Surgery, Petersfield Medical Practice, Prior's Field Surgery, Queen Edith's Medical Practice, Queen Street Surgery, Rainbow Surgery, Ramsey Health Centre, Riverside Practice, Roman Gate Surgery, Rosalind Franklin House, South Street Surgery, Thaxted Surgery, The Health Centre (Bury St Edmunds), The Old Exchange, The Surgery Stanground, Townley Close Health Centre, Trumpington Street Medical Practice, Werrington Health Centre and York Street Medical Practice.

**Contributors** ML, ECFW, CEB and SJG designed the concept for the paper. ML performed the statistical analysis, interpreted the data and drafted the manuscript. SJG and CEB were involved in collecting the data. ECFW provided statistical support. All authors critically revised the intellectual content of the manuscript and approved its final version.

**Funding** ADDITION-Cambridge was supported by the Wellcome Trust (grant reference No G061895), the Medical Research Council (grant reference no: G0001164), National Health Service R&D support funding (including the Primary Care Research and Diabetes Research Networks), and the National Institute for Health Research. We received an unrestricted grant from University of Aarhus, Denmark, to support the ADDITION-Cambridge trial. Bio-Rad provided equipment to undertake capillary glucose screening by HbA1c in general practice. SJG is a National Institute for Health Research (NIHR) Senior Investigator. The Primary Care Unit is supported by NIHR Research funds. SJG received support from the Department of Health NIHR Programme Grant funding scheme (RP-PG-0606-1259). ECFW is funded by the NIHR Cambridge Biomedical Research Centre.

**Disclaimer** The views expressed in this publication are those of the authors and not necessarily those of the NHS, the NIHR or the Department of Health.

**Competing interests** None declared.

**Patient consent** Obtained.

**Ethics approval** Ethical approval was granted by the Eastern Multi-Regional Ethics Committee- UK (ref 02/5/54).

**Provenance and peer review** Not commissioned; externally peer reviewed.

**Data sharing statement** The access policy for sharing is based on the MRC Policy and Guidance on Sharing of Research Data from Population and Patient Studies. All data sharing must meet the terms of existing participants' consent and study ethical approvals. Information on data and data requests can be found on http://epi-meta.medschl.cam.ac.uk/includes/addcam/addcam.html. In case of questions please contact datasharing@mrc-epid.

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
