## [Reviewer comments · BMJ Open]

ARTICLE DETAILS

TITLE (PROVISIONAL)	How good are GPs at adhering to a pragmatic trial protocol in primary care? Results from the ADDITION-Cambridge cluster-randomized pragmatic trial
AUTHORS	Laxy, Michael; Wilson, Edward; Boothby, Clare; Griffin, Simon

VERSION 1 - REVIEW

REVIEWER	Simeon Pierre Choukem Faculty of Health Sciences, University of Buea Cameroon
REVIEW RETURNED	31-Dec-2016

GENERAL COMMENTS	General comments In this paper, Laxy M et al have explored the adherence of general practitioners to protocols of a pragmatic trial that compared the outcome of intensively treated vs routine care in screen-detected type 2 diabetes in Cambridge. The results are of great interest for clinical trials in general and pragmatic trials specifically, because they raise the awareness on aspects of trials that can significantly influence the outcomes and the conclusions, but are often overlooked. There are however some minor issues to address before it can be accepted for publication. Specific comments Methods - Page 6, 2nd paragraph: There seems to be confusion between surgeries' selection and patients' selection. Though the selection of the 63 patients with 1ary endpoint is clear, it is not understandable how you arrived to 126 additional patients without 1ary endpoint. Was it two additional patients per surgery? In which case it should be 2 x 49 surgeries = 98. Authors should make it simple and clear.- Figure 1: "63 practices agreed and assessed for eligibility"- Though surgeries and practices may be used interchangeably, I suggest you use the same term consistently in the methods and figure 1.- Table 1: Use "total cholesterol" Results - Table 2: author should use "HDL cholesterol"
--

REVIEWER	Dr Andrew P Dickens University of Birmingham United Kingdom
REVIEW RETURNED	26-Feb-2017

GENERAL COMMENTS	I was very interested to read this paper and the novel approach they used to assess GP adherence to a pragmatic trial protocol. Data was presented clearly with appropriate, balanced discussion regarding the data itself and implications for future research. I would recommend this paper to be accepted with minor revisions addressing the below points, the vast majority of which are simply asking for clarification. Methods: Page 6, line 18:  • The description of the 1:2 ratio of those with and without primary endpoints could be written more succinctly. • The authors should provide justification for why this ratio was used Page 6, line 24:  • It could be made clearer that this sentence is describing the group allocation of the selected patients in the previous sentence. Page 6, line 38:  • Were there 3 extras GP and nurse consultations in each of the first 3 years after diagnosis, or were they spread across the 3 year period? Page 7, line 49:  • Is it worth including details of the electronic searches (e.g. Read codes) as an appendix? This could be overly-complex for the authors, but if it is possible it would help readers replicate the study in future Page 8, line 12:  • Were the contacts with GPs/nurses restricted to diabetes-related consultations, or all consultations irrespective of reason for attendance? Page 9, line 37:  • Replace 'multiply-imputed' with 'multiple-imputed' Results: Page 10, page 57:  • The authors state no trend was observable. Although that is true for nurse consultations, the IT group seemed to have consistently higher GP consultations, albeit not significantly different from the RC group Page 11, 2nd paragraph:  • It may be my misunderstanding, but shouldn't this paragraph reflect the time interaction whereby only glucose lowering medications were significantly different between groups? Page 12, line 44  • The authors should perhaps comment on the relative high proportion of patients receiving unnecessary BP lowering/ACDE
--

	inhibiting drugs in RC group, compared with IT Page 14, line 48:  • The authors make an important point about the financial payments perhaps only being an incentive for participation, rather than the intended resource for additional consultations. Within primary care research, offering practices financial reimbursement for additional tasks is often essential. Had a mixed methods approach been adopted, it would have been interesting to explore the reason for no increase in consultations e.g. current frequency perceived to be sufficient, treatment algorithms seen to be the main element of the intervention etc  Page 15, line 25:  • "...there BP levels..." should be "...their BP levels..."  Strengths & limitations: Page 18, 2nd paragraph:  • While data from electronic records was able to provide significant information, the authors acknowledge the limitation that it does not capture the shared decisions of the GP and patient that inevitably contributed to treatment decisions. The authors should also acknowledge that their data cannot inform whether information was handed out to patients; this is presented as a component of the intensive treatment described on page 7, but is currently not mentioned in the paper. 
--	--

REVIEWER	Andrea Siebenhofer Institute of General Practice and Evidence-based Health Services Research; Medical University Graz - Austria Institute for General Practice; Goethe University Frankfurt am Main - Germany
REVIEW RETURNED	27-Feb-2017

GENERAL COMMENTS	As a clinical researcher performing pragmatic trials myself and knowing how difficult it is to carry out high quality trials, I thought this manuscript was very interesting. The findings described in the manuscript seem to me to be very valuable, especially as knowledge about adherence to long term pragmatic trial protocols and the potential barriers involved should help researchers realise the importance of prospective considerations and planning carefully when designing new study protocols. The manuscript describes general practitioner (GP) adherence to the long-term pragmatic trial protocol of the ADDITION-Cambridge trial by comparing intensive vs. routine care in a subsample of 189 screen-detected diabetes patients. There was no statistical difference between treatment groups in terms of annual contacts with GPs and nurses, but IT patients received more glucose-, hypertensive- and lipid-lowering drugs. Introduction and methods: - Well written introduction and method section Methods: Intensive treatment vs. routine care: Am I correct in understanding that glucose treatment was initiated when HbA1c was $\geq 6.5\%$, but
--

	that the treatment target was < 7% (table 1 in the protocol paper in the BMC Public Health 2009), and that there is also a difference in treatment recommendation thresholds and target levels with regard to lowering cholesterol levels and relieving hypertension? Are you aware of any other studies that differentiate between treatment targets and thresholds at which treatment is recommended? Results Can you explain why only 173 patients were analysed? How did it come about, as it was a post-hoc analysis that started with 62 patients with a primary outcome, and two further randomly selected patients from the same surgeries. Table 2 Baseline characteristics: Was there already a primary endpoint for the patients at the beginning of the trial? Please clarify this? Discussion: I like the discussion and the list of potential reasons for the lack of differences between the treatment groups. This is also of special interest when one considers that there are many possible reasons for failing to achieve significant results in studies involving complex interventions. The more papers are published that make recommendations on how to plan future pragmatic trials, the more researchers will be aware of potential pitfalls.
--	---

VERSION 1 – AUTHOR RESPONSE

Reviewer: 1

Reviewer Name: Simeon Pierre Choukem

Institution and Country: Faculty of Health Sciences, University of Buea, Cameroon Please state any competing interests or state 'None declared': None declared

Please leave your comments for the authors below Reviewer's report

Title: How good are GPs at adhering to a pragmatic trial protocol in primary care? Results from the ADDITION-Cambridge cluster-randomized pragmatic trial

General comments

In this paper, Laxy M et al have explored the adherence of general practitioners to protocols of a pragmatic trial that compared the outcome of intensively treated vs routine care in screen-detected type 2 diabetes in Cambridge. The results are of great interest for clinical trials in general and pragmatic trials specifically, because they raise the awareness on aspects of trials that can significantly influence the outcomes and the conclusions, but are often overlooked. There are however some minor issues to address before it can be accepted for publication.

Specific comments

Methods

- Page 6, 2nd paragraph: There seems to be confusion between surgeries' selection and patients' selection. Though the selection of the 63 patients with 1ary endpoint is clear, it is not understandable how you arrived to 126 additional patients without 1ary endpoint. Was it two additional patients per surgery? In which case it should be 2 x 49 surgeries = 98. Authors should make it simple and clear. We agree that this section was a little difficult to follow. We have therefore rewritten this section completely and added a justification for our approach.

- Figure 1: "63 practices agreed and assessed for eligibility"
We corrected this typo.

- Though surgeries and practices may be used interchangeably, I suggest you use the same term consistently in the methods and figure 1.

We now use the term 'GP surgery' throughout the manuscript.

- Table 1: Use "total cholesterol"

Done.

- Results

- Table 2: author should use "HDL cholesterol"

Done.

Reviewer: 2

Reviewer Name: Dr Andrew P Dickens

Institution and Country: University of Birmingham, United Kingdom Please state any competing interests or state 'None declared': None declared

Please leave your comments for the authors below I was very interested to read this paper and the novel approach they used to assess GP adherence to a pragmatic trial protocol. Data was presented clearly with appropriate, balanced discussion regarding the data itself and implications for future research. I would recommend this paper to be accepted with minor revisions addressing the below points, the vast majority of which are simply asking for clarification.

Methods:

Page 6, line 18:

- The description of the 1:2 ratio of those with and without primary endpoints could be written more succinctly.

We agree that this section was a little difficult to follow. As per reviewer 1's comments we have rewritten this section.

- The authors should provide justification for why this ratio was used

The reason for this approach is mainly one of efficiency. Assessment, extraction and preparation of data from electronic patient medical records from different GP surgeries and software systems is time consuming and expensive. In light of this context in the planning phase of the trial this rather pragmatic approach (all participants with a primary endpoint + 2 'controls') was defined. We added two sentences to the methods section to clarify this for the reader.

Page 6, line 24:

- It could be made clearer that this sentence is describing the group allocation of the selected patients in the previous sentence.

We have revised the wording to clarify the connectedness of this sentence with the previous (rewritten) section.

- Page 6, line 38:

Were there 3 extra GP and nurse consultations in each of the first 3 years after diagnosis, or were they spread across the 3 year period?

- There were 3 extra consultations per year. We have revised the wording to make this clear to the reader.

Page 7, line 49:

- Is it worth including details of the electronic searches (e.g. Read codes) as an appendix? This could

be overly-complex for the authors, but if it is possible it would help readers replicate the study in future. In principle this is a good idea. However, we doubt that the effort required to prepare this for the readers would pay off. The software systems of the GP surgeries were not uniform and multiple (probably unique) steps were necessary to extract the data. Also the preparation of data in Excel and SAS required a series of specific steps which are closely related to the given structure/format of the extracted data. The series of manual and automatic (SAS, excel) steps that are necessary therefore highly depend on the specific research question. As the timely effort to prepare and describe these multiple steps is significant and the added value for the readers is expected to be small, we decided to not include these information in the appendix. Furthermore, one of the GP data systems (MYQUEST) is no longer available.

Page 8, line 12:

- Were the contacts with GPs/nurses restricted to diabetes-related consultations, or all consultations irrespective of reason for attendance?

We were unable to distinguish between diabetes-related and diabetes unrelated consultations. We revised this sentence to clarify this issue.

Page 9, line 37:

- Replace 'multiply-imputed' with 'multiple-imputed'
- Done.

Results:

Page 10, page 57:

- The authors state no trend was observable. Although that is true for nurse consultations, the IT group seemed to have consistently higher GP consultations, albeit not significantly different from the RC group.

This is correct. However, this (constant) difference in the number of GP contacts between the trial arms did not approach statistical significance and we are therefore reluctant to describe this a 'trend'. The term 'trend' in this sentence referred to the fact that there was no significant increase or decrease in the number of contacts over time. We rephrased this sentence to make this clear.

Page 11, 2nd paragraph:

- It may be my misunderstanding, but shouldn't this paragraph reflect the time interaction whereby only glucose lowering medications were significantly different between groups?

The analysis we made in this paper differs from a classical mixed model approach which is often used in the analysis of RCTs. In such classical models, both baseline and follow-up measures are utilized and the interaction $\text{intervention} \times \text{time}$ is the estimate of interest (effect of the intervention). We do not have information about the treatment at baseline (treatment before randomization/start of study). The main effect of the applied mixed model in this study describes therefore the average difference in treatment intensity between the RC and IT arm over the follow-up time and the interaction $\text{intervention} \times \text{time}$ describes if there occur any changes in the difference of treatment intensity over time (f.ex. due to changes of guidelines in the routine care group).

Page 12, line 44

- The authors should perhaps comment on the relative high proportion of patients receiving unnecessary BP lowering/ACE inhibiting drugs in RC group, compared with IT

Thank you for this valuable remark. We want to emphasize that these prescriptions might not be unnecessary – it could also be the case that there is a large proportion of well controlled patients. We discuss this point in the discussion section 3 under "principle findings". As we do not want to mix up results with interpretations we now added the following sentence to the result section. "Of note, a large proportion of patients in the RC arm with BP levels below the threshold were prescribed BP

lowering medication.”

Page 14, line 48:

- The authors make an important point about the financial payments perhaps only being an incentive for participation, rather than the intended resource for additional consultations. Within primary care research, offering practices financial reimbursement for additional tasks is often essential. Had a mixed methods approach been adopted, it would have been interesting to explore the reason for no increase in consultations e.g. current frequency perceived to be sufficient, treatment algorithms seen to be the main element of the intervention etc

We agree very much that a mixed method approach with some qualitative data would have been a very valuable extension to this study. Unfortunately, we did not collect such data and must infer our conclusions from the quantitative analysis of the electronic primary care records. The ADDITION study itself did include qualitative work with practitioners, but this focused mainly on the screening phase of the study (Fam Pract 2010;27:386-94).

Page 15, line 25:

- “...there BP levels...” should be “...their BP levels...”

We corrected this typo.

Strengths & limitations:

Page 18, 2nd paragraph:

- While data from electronic records was able to provide significant information, the authors acknowledge the limitation that it does not capture the shared decisions of the GP and patient that inevitably contributed to treatment decisions. The authors should also acknowledge that their data cannot inform whether information was handed out to patients; this is presented as a component of the intensive treatment described on page 7, but is currently not mentioned in the paper. This is a very good point. Handing out information material was part of the intervention. However, in our data source we had no information concerning whether this was done by the GPs. We added a sentence relating to this to the limitations section.

Reviewer: 3

Reviewer Name: Andrea Siebenhofer

Institution and Country: Institute of General Practice and Evidence-based Health Services Research; Medical University Graz - Austria; Institute for General Practice; Goethe University Frankfurt am Main - Germany Please state any competing interests or state 'None declared': None declared

Please leave your comments for the authors below As a clinical researcher performing pragmatic trials myself and knowing how difficult it is to carry out high quality trials, I thought this manuscript was very interesting. The findings described in the manuscript seem to me to be very valuable, especially as knowledge about adherence to long term pragmatic trial protocols and the potential barriers involved should help researchers realise the importance of prospective considerations and planning carefully when designing new study protocols.

The manuscript describes general practitioner (GP) adherence to the long-term pragmatic trial protocol of the ADDITION-Cambridge trial by comparing intensive vs. routine care in a subsample of 189 screen-detected diabetes patients. There was no statistical difference between treatment groups in terms of annual contacts with GPs and nurses, but IT patients received more glucose-, hypertensive- and lipid-lowering drugs.

Introduction and methods: Well written introduction and method section

Methods:

- Intensive treatment vs. routine care: Am I correct in understanding that glucose treatment was initiated when HbA1c was $\geq 6.5\%$, but that the treatment target was $< 7\%$ (table 1 in the protocol paper in the BMC Public Health 2009), and that there is also a difference in treatment recommendation thresholds and target levels with regard to lowering cholesterol levels and relieving hypertension? Are you aware of any other studies that differentiate between treatment targets and thresholds at which treatment is recommended?

Yes this is correct. There is a difference between treatment thresholds and target levels. Part of the rationale for this approach was to counter the phenomenon of ‘clinical inertia’ in which patients and practitioners observe the level of a risk factor rising but put off increasing treatment of the risk factor. This has been well-described for treatment of hyperglycaemia (Khunti K et al. Diabetes Care 2013;36:3411-7). Disaggregating the threshold for recommending treatment from the therapeutic target for the risk factor being treated is seen in other contexts. For example, the use of statins, which are recommended based on modelled cardiovascular risk but are then monitored and adjustments made to dosage based on LDL or total cholesterol levels.

Results

- Can you explain why only 173 patients were analysed? How did it come about, as it was a post-hoc analysis that started with 62 patients with a primary outcome, and two further randomly selected patients from the same surgeries.

The analysis sample was reduced from $n=189$ to $n=173$ as for 16 participants we simply had no usable information from the electronic primary care records. The reason for this is that some participants changed their GP or that sometimes the data were not accessible. The shrinkage of sample size from 189 to 173 is explained in the first section of “statistical analysis”.

- Table 2 Baseline characteristics: Was there already a primary endpoint for the patients at the beginning of the trial? Please clarify this?

No, these numbers describe how many patients in the analysis sample experienced a primary endpoint over the 5 years of follow-up. As this information does not really provide valuable information to the reader we omitted this rather misleading piece of information from the table.

Discussion:

- I like the discussion and the list of potential reasons for the lack of differences between the treatment groups. This is also of special interest when one considers that there are many possible reasons for failing to achieve significant results in studies involving complex interventions. The more papers are published that make recommendations on how to plan future pragmatic trials, the more researchers will be aware of potential pitfalls.

We thank the reviewer for this comment.

VERSION 2 – REVIEW

REVIEWER	Simeon Pierre Choukem Faculty of Health Sciences, University of Buea
REVIEW RETURNED	29-Apr-2017

GENERAL COMMENTS	No additional comment
-----------------------

REVIEWER	Dr Andrew P Dickens University of Birmingham, United Kingdom
REVIEW RETURNED	15-May-2017

GENERAL COMMENTS	I agree with the vast majority of revisions made, and my outstanding comments are very minor indeed. I have only specified 'minor revision' in case the editor considers it useful to add the few sentences I suggest below. Author's response to original comment: Page 7, line 49: The author's response is possibly true regarding balance of time vs benefit for the reader. However, as an aside Miquet is a tool for extracting data from all clinical systems, rather than being a clinical system itself. Page 10, line 57: I think the revision makes it less clear than the original. Perhaps revert to the original text, but replace "...and no consistent trend over time" with "...and no statistically significant trend over time" Would this be clearer? Page 11, 2nd para: The authors' response is fair, but would it not have been possible to extract treatment at baseline as part of the data extraction process? This could have made the analyses more robust. Page 14, line 48: Perhaps it would be worth adding a sentence to the Discussion to make this point?
--

REVIEWER	Andrea Siebenhofer Institute of General Practice and Evidence-based Health Services Research / Medical University Graz / Austria Institute of General Practice / Goethe University / Germany
REVIEW RETURNED	11-May-2017

GENERAL COMMENTS	There are no further comments and from my point of view it can be accepted.
---

VERSION 2 – AUTHOR RESPONSE

Reviewer: 2

Reviewer Name: Dr Andrew P Dickens

Institution and Country: University of Birmingham, United Kingdom Please state any competing interests or state 'None declared': None declared

Please leave your comments for the authors below I agree with the vast majority of revisions made, and my outstanding comments are very minor indeed. I have only specified 'minor revision' in case the editor considers it useful to add the few sentences I suggest below.

Author's response to original comment:

Page 7, line 49: The author's response is possibly true regarding balance of time vs benefit for the reader. However, as an aside Miquet is a tool for extracting data from all clinical systems, rather than being a clinical system itself.

This is correct. Thank you for this comment.

Page 10, line 57: I think the revision makes it less clear than the original. Perhaps revert to the original text, but replace "...and no consistent trend over time" with "...and no statistically significant trend over time" Would this be clearer?

This is a good suggestion. We changed the wording again and the sentence reads now as follows:

“We found no difference in the mean annual number of contacts with GPs (IT: 5.80, vs. RC: 5.15, $\beta=0.65$ [95%-CI: -0.95, +2.26]) or nurses (IT: 5.34 vs. RC: 5.49, $\beta = -0.15$ [-1.77, +1.48]) and no statistically significant trend over time.”

Page 11, 2nd para: The authors' response is fair, but would it not have been possible to extract treatment at baseline as part of the data extraction process? This could have made the analyses more robust.

In hindsight we agree this would have made the analyses more robust and in future we will ensure such data are extracted given the opportunity. In mitigation, we feel that in this case it would have made little difference: the screen-detected participants in the trial were not prescribed glucose-lowering treatment at or prior to baseline. In addition, we know that the groups were broadly comparable at baseline in terms of their self-reported blood pressure lowering or lipid-lowering medication (Griffin et al. Lancet 2011; 378: 156–6, Table 3).

Page 14, line 48: Perhaps it would be worth adding a sentence to the Discussion to make this point? We added a sentence to following sentence to this section: “Qualitative interviews with the GPs about their perspectives on the intervention, as conducted in the screening phase of the ADDITION study [28], would have been a valuable add on to address this question.”